# Effects of Boundary Condition Models on the Seismic Responses of a Container Crane

**Jungwon Huh** [1] , **Van Bac Nguyen** [1] , **Quang Huy Tran** [1,2] , **Jin-Hee Ahn** [3] **and Choonghyun Kang** [1,*]

1   Department of Civil and Environmental Engineering, Chonnam National University, Yeosu, Chonnam-do 59626, Korea; jwonhuh@chonnam.ac.kr (J.H.); 186197@jnu.ac.kr (V.B.N.); 157042@live.jnu.ac.kr (Q.H.T.)
2   Department of Civil Engineering, Nha Trang University, Khanh Hoa 57000, Vietnam
3   Department of Civil Engineering, Gyeongnam National University of Science and Technology, Jinju, Gyeongnam 52725, Korea; jhahn@gntech.ac.kr
*   Correspondence: kangcivil@gmail.com; Tel.: +82-10-6357-8142



**Featured Application: This study provides crane designers and experts with a comprehensive view of the effect of boundary conditions to response of container cranes subjected to an earthquake event. These effects were analyzed in detail in terms of movement of legs, portal drift, reaction of legs, and total base shear.**

**Abstract:** In recent years, several large earthquakes have caused the collapse of container cranes, which have resulted in halting of freighting, and significantly affected the economy. Some reports are concerned the uplift and derailment events of crane legs, and the collapse of the crane itself. In this study, the effects of different boundary conditions used in the numerical method are investigated for a container crane under seismic excitation. Three different boundary conditions are considered in terms of the connection of the crane's legs (wheels) and the ground (rails), namely pin support (PIN), gap element (GAP), and Friction contact (FC) elements, by using the SAP2000 program for a typical container crane. Then, time history dynamic analyses are conducted using nine recorded ground motions. Dynamic behaviors of the container crane are studied in terms of the total base shear, portal drift, and relative displacement of legs, by investigating the three types of base boundary conditions. The results of the study show that when the intensity of earthquakes is large enough to create uplift and derailment events, the selection of the boundary condition model considerably affects the dynamic responses of the container crane. In addition, when uplift and derailment of the crane occur, the FC support condition is the most compatible with the real behavior of the crane. On the other hand, under low seismic excitation, there is no significant difference of the crane behavior according to the choice of boundary condition model.

**Keywords:** container crane; friction contact; boundary condition; portal drift; total base shear; uplift; derailment; ground motion; spectral acceleration; time history analysis

## 1. Introduction

The impact of an earthquake on civil structures (i.e., buildings, bridges, dams, tunnels, and container cranes) is always most concerning for engineers, because it badly affects people and the economy. The container crane is a steel structure built at seaports for loading and unloading container ships. It is an important link for transporting freight between countries, so it also receives the special attention of countries having seaports. However, container cranes are one of the most vulnerable

elements of the port system, and are often damaged under moderate seismic loads. Chang's research [1] into disasters of the Port of Kobe after the 1995 earthquake showed that although in terms of magnitude ($M_w$ = 6.9) it was a moderate earthquake, it caused an estimated US$10 billion loss (at exchange rates at the time), because the Port was essentially shut down, and the scale of damage was so large that repairs were not completed until March 1997 (over two years after the earthquake). This means that the traffic that would have gone through Kobe was diverted to other ports. Moreover, modern container cranes are larger than ever before to satisfy the demand for bigger ships, so the cranes can easily become unstable under seismic loads. For these reasons, evaluating the dynamic response of container cranes under seismic excitation needs to be studied further, to ensure their structural stability.

Investigating the response of the container crane under seismic load has often been researched by experimental and numerical methods. Jacobs et al. [2] used a shake table experiment that was performed on a 1:20 scale model to characterize the uplift behavior, then compared that with the 2-D and 3-D finite element models. The authors showed that the simulated model is sufficient to perform dynamic time history analysis for evaluating the overall behavior, including uplift and derailment phenomena. Azeloglu et al. [3] investigated the seismic behavior of the container crane by shake table test (using a 1:20 scaled container crane) and a mathematical model, and also concluded that the dynamic behavior, including in the time and frequency domains, of the crane structure could be represented by the mathematical model. Kanayama et al. [4] carried out studies on the dynamic response of cranes by a series of shaking table tests (using 1/25 and 1/8 scaled crane models). In their research, the recoded ground motions were applied critically in the trolley-boom direction to find components that were the most destructive. The results showed that the derailment of the wheels happens often under large seismic excitations. The shake table test is an effective method to evaluate the dynamic response of the container crane under seismic excitation. However, it is not an economical method, and sometimes it may not be necessary to conduct it in small projects. In some case, the numerical method using commercial software will produce a reliable result, and is more economical.

All components of a container crane are supported and moved via its wheels. In normal conditions, the crane still remains under its gravity load, which means that the wheels are fixed to the rail or ground, so that the boundary condition in a simulated model could be a pin support that can resist both vertical and horizontal forces. On the other hand, if the seismic load is large, the wheels might be uplifted. In particular, previous research in Japan has proven that the uplift and derailment of cranes can be reached under just moderate earthquakes [4–6]. In this case, the base shear of the uplifted wheels is lost, and the uplifted wheel can move freely in the vertical and horizontal directions, so the pin support is not suitable to be applied. For dealing with this problem, Kanayama et al. [4] proposed two models: a planar two-dimensional and a three-dimensional model for evaluating the uplift and derailment events. These behaviors were simulated by frictional contact elements programmed by OpenSees [7], in which the leg base works as pin support, until it uplifts when the coefficient of friction is sufficiently high. Based on the study of Kosbab et al. [8], they proposed the friction coefficient of 0.75, in order to evaluate the seismic response of a modern jumbo container crane. In Kosbab et al.'s experimental scale models, they also concentrated on treating the boundary condition, in which a friction-enhancing disk was placed between the bottom of the column and the top of the steel base plate on the shake table. The coefficient of friction between the disk and steel plate was in the range of 0.6 to 0.8. Another simpler link that uses an element was developed to model the dynamic impact of two nodes; this link is called an elastic-no-tension element. The element type allows only the generation of compressive forces, and movement in one direction. This element was proposed for a link between crane legs and rail by Liftech Consultants Inc. [9] to analyze the capacity of the non-uplifted side of the container crane. The report of Miyata et al. [10] on the seismic performance-based design method for container cranes also suggested the use of this element to simulate the link of crane legs and ground.

This study investigated the effects of boundary condition models on the dynamic response of the container crane under seismic excitation by time history analysis. Nine horizontal response spectra of actual earthquakes are adjusted by scaling the amplitude of each response spectrum to a specific

target response spectrum acceleration at the fundamental period of the container crane. Ten levels of target spectral acceleration are determined for evaluating the seismic response of the container crane. Some response characteristics, i.e., reaction of legs, portal drift, and movement of legs, are investigated by time history analysis. The different responses of the structure due to different types of boundary condition models are discussed as well.

## 2. Description of a Numerical Model of the Container Crane

In this study, a 3D finite element model (FEM) of the container crane is generated by SAP2000. Figure 1 shows that the FEM uses the real dimensions of the Korean container crane. The container crane has crucial dimensions of: total height of 77.8 m from the ground to the top of the container crane; length from the end of the trolley girder to the end of the boom of around 136 m; the portal frame that controls the critical portal sway mode has a width of 30.5 m and its height is 17.5 m; it has an outreach of 63 m, and a back reach of 20.0 m. Most of the structural components are made of built-up stiffened box sections, except for diagonal braces that were made of tube sections; the forestay and backstay are made of wide-flange shapes. The properties of the material complied with the Japanese industrial standards (JIS) JIS-SM490Y and JIS-STK490. The FEM is composed of 9916 elements; 9912 frame elements, and 4 link elements that are used to simulate the connection of ground and structure. The moment in the end and beginning of elements, i.e., the forestay, backstays, and diagonal braces, are released; thus, these elements work as truss elements. All non-structural loads, i.e., machinery house, drive trucks, stairs, stowed pins, snag device, and boom hoist rope, are simulated as concentrated or distributed loads. The total weight of the container crane is 13,883 kN.

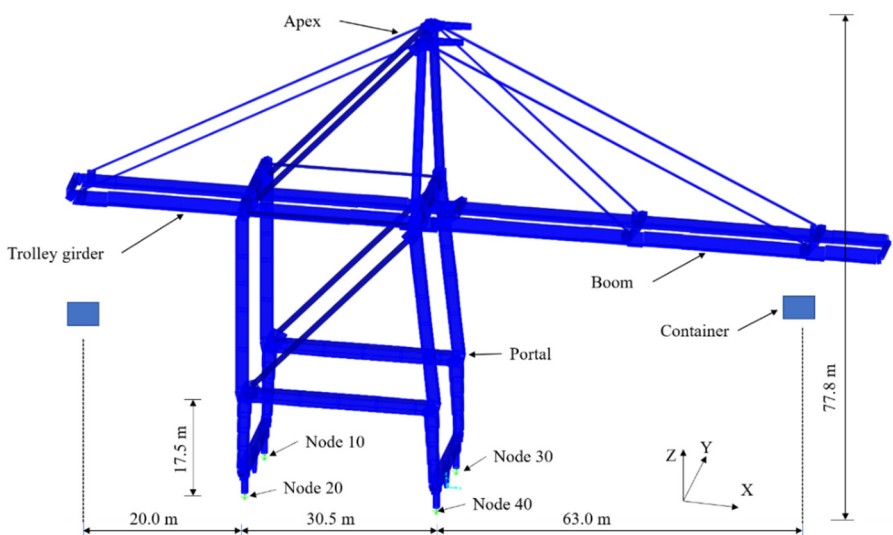

**Figure 1.** 3D numerical model of the Korean container crane.

### 2.1. Modeling of Boundary Conditions

As mentioned above, the dynamic response of container cranes with uplift and derailment events is a complicated event. In this study, three simulated models of the base boundary conditions are investigated by SAP2000 to evaluate their effects on the response of the container crane. First, the normal boundary condition idealization is to use a simple pin (PIN), so that the container crane's legs are in bonded contact with the ground (rail). In this case, neither uplift or derailment event is permitted to develop, so it is appropriate for dynamic analyses, in which the level of load does not create an uplift and derailment event for the container crane. In fact, there are cases of analysis when the uplift response is not essential, i.e., for static analysis (gravity, service, etc.) where the uplift event rarely happens, for dynamic analysis of cranes located in regions of low seismicity, and for cranes that tie down in harbor because of large wind loads, and where therefore, PIN is sufficient. When a

PIN support is applied, a simple check should be done to ensure that all vertical reactions of supports throughout the analysis remain compressive.

Second, the elastic-no-tension (ENT) element simulated by a link between the structure and the ground was recommended by some previous studies, such as Tran et al. [11,12], who when analyzing a 3D model of a container crane by the time history method, and used a link element in SAP2000 for simulating a connection between container crane and crane rail. By using this element, the compressive vertical reactions of legs may be developed under general load. However, when the dynamic load makes a vertical reaction to reduce to zero, no tensile reaction developed. This means that the ENT element allows the legs to move freely in a vertical direction (allowing an uplift event). The weakness of this link is that it does not allow a horizontal movement of legs during uplift events. Therefore, the derailment event is impossible to develop as a result of the leg base being continuously fixed in a horizontal direction despite uplifting, so the leg base moves perfectly in a vertical direction. Consequently, during an uplift event the horizontal reaction is developed, while the vertical reaction reduces to zero. This means that the reaction will redistribute the internal forces and the deformation of the structure, so that the uplifted leg can carry some portion of base shear. In this study, the elastic-no-tension element is defined by the GAP element in SAP2000. Equation (1) gives the nonlinear force and deformation relationship of the support:

$$F = \begin{cases} k(d + open) & if\ d + open < 0 \\ 0 & otherwise \end{cases} \tag{1}$$

where $k$ is the spring constant, and "open" is an initial gap opening that must be zero or positive (open = 0 for the link of legs of container crane and ground). All internal deformations are independent, which means that the opening of a gap for one deformation does not affect the behavior of the other deformations. Equation (1) shows that this approach is relatively straightforward, so the energy loss during a collision cannot be modeled. In this study, the stiff spring constant ($k$) was recommended to be 43,781 kN/cm (25,000 kips/in) [13], which was also shown to be insensitive to changes of magnitudes [14,15]. Figure 2b shows the model of GAP.

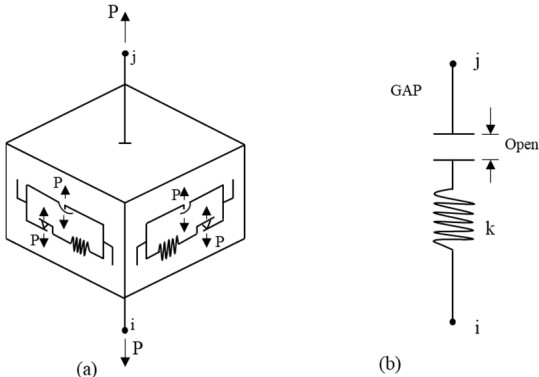

**Figure 2.** Boundary condition in SAP2000: (**a**) FC element, (**b**) GAP element.

Third, the frictional contact (FC) boundary element relates the horizontal force capacity to the compressive vertical force, as can be seen in the Mohr-Coulomb Equation (2):

$$F_{H,max} = \mu.F_v \tag{2}$$

In the above equation, the largest horizontal force that can resist a slip of legs is equal to the frictional coefficient ($\mu$) multiplied by the vertical reaction. Here, the frictional coefficient ignores the difference between static and kinetic friction. When using this FC element for simulating the boundary of the container crane, both the horizontal and vertical reaction of legs will reach zero if

the legs have uplift and derailment events. In addition, the horizontal and vertical movement of legs could be captured in uplift and derailment events. These special behaviors of the link could be modelled by commercial software, such as ABAQUS, OpenSees, ANSYS, and SAP2000, and are simulated by an elastomeric bearing as a two-node discrete element with stiffness in each of the six principal directions (three translations and three rotations), represented by linear or nonlinear springs between two nodes. Analytical expressions for force and stiffness can be used to define a spring in any direction. In SAP2000, the friction isolator, as shown in Figure 2a, allows compression-only property in the axial direction, similar to a GAP element as mentioned above, and has coupled frictional behavior in the two horizontal shear directions. The model using the base shearing behavior was introduced by Park et al. [16], and was then studied for base-isolated structures by Nagarajaiha et al. [17]. This element could also be used to model gap and friction behavior between contacting surfaces by taking into account efficient numbers for the stiff spring constant and damping coefficient, so the uplift and derailment events of wheels could be obtained in the simulated model.

This FC boundary element was used by Kosbab [18] in the OpenSees platform, and also compared to an experimental model. As a result, the author recommended using the FC boundary element to capture the critical uplift and derailment responses in a realistic but simplified manner. Jaradat et al. [19] used this type of element to simulate the interaction of crane–wharf in nonlinear time history analysis by SAP2000. They recommended the stiffness for sliding and uplift to be eight times the gravity reaction, and assumed the coefficient of friction to be 0.8, and the damping 5%. In our study, the properties of friction isolators in SAP2000 were chosen to exactly capture the uplift and derailment events of the container crane as follows: the stiff spring constant ($k_k$) is assumed to be 43,781 kN/cm (25,000 kips/in), and the damping coefficient ($C_k$) is based on the coefficient of restitution (e) by an equation of loss energy during impact, as proposed by Muthukumar [13] and shown in Equations (3) and (4). These parameters have also been shown to not be very sensitive to changes of magnitude for simulating collisions [14,15]. The coefficient of friction is suggested to be 0.6 to 0.9 by experimental model in the research of Kosbab and Jacobs. If the parameter is large enough, the simulated model of the leg base acts as essentially pinned until it uplifts; if a suitable parameter is used to capture the uplift and derailment event, at that point the simulated support behaves freely. In this study, the coefficient of friction was assigned to be 0.75. The damping coefficient is calculated by the following equation:

$$\zeta = -\frac{ln\ e}{\sqrt{\pi^2 + (lne)^2}} \tag{3}$$

$$C_k = 2\zeta\sqrt{k_k m} \tag{4}$$

where $m$ is the structural mass contributing to dynamic vibration of the stiff spring, it was considered for each leg of the crane ($m$ = 3470 kN); $\zeta$ is the damping ratio; and $e$ is the coefficient of restitution, which contributes as a dashpot constant of impact elements to determine the amount of energy dissipated during impact. If two masses collide each other with arbitrary velocities, it is easy to express the dashpot constant of the impact element in terms of the $e$ by equating the energy losses during impact. The elastic impact ($e$ = 1.0) corresponds to a damping ratio $\zeta$ = 0, and the completely plastic impact ($e$ = 0) corresponds to $\zeta$ = 1.0. In this study, $e$ = 0.6 was assumed, as recommended by Muthukumar [13], corresponding to a damping ratio of $\zeta$ = 0.16.

*2.2. Dynamic Analysis of the Container Crane*

In this study, the time history analysis (THA) method is selected to determine the dynamic response of the container crane under seismic excitation. The fundamental dynamic equilibrium equations of THA can be expressed as Equation (5) by SAP2000 [20,21]:

$$\mathbf{K}\boldsymbol{u}(t) + \mathbf{C}\dot{\boldsymbol{u}}(t) + \mathbf{M}\ddot{\boldsymbol{u}}(t) = \boldsymbol{r}(t) \tag{5}$$

where **K**, **C**, and **M** are the stiffness matrix, damping matrix, and mass matrix, respectively; **u**, $\dot{u}$, and $\ddot{u}$ are the displacements, velocities, and accelerations of the structure; and *r* is the applied load.

There are several options for solving Equation (5), i.e., linear and nonlinear, modal, and direct integration. In this study, the intensity of recorded ground motion is increased to evaluate the uplift and derailment events of the container crane. In this study, the maximum mean of acceleration spectra of ground motions ($S_a = 0.9$ g) is much larger than the elastic acceleration spectra, according to KDS 2016; as a result, the dynamic response of the container crane is expected to deform past the limit point for elastic behavior. Therefore, the nonlinear analysis and direct-integration solution are applied to solve Equation (5). In addition, the properties of the plastic hinges are defined in ASCE/SEI 41-13 [22]. The plastic hinge mechanism is a model that is appropriate for the maximum moment that occurs at the end of members [23,24], and beam-hinge and column-hinge mode were assigned for the portal frame and legs, as in Figure 3a. Assigning plastic hinges has been found to be suitable in some previous research, such as that recommended in PIANC [25]; in fact, in the 1995 Kobe earthquake, the extensive damage to the container crane was local buckling of the legs and derailment [26].

The fundamental period of the Korean container crane was analyzed by the Ritz vectors method. Twelve natural frequencies of the main vibration modal characteristics were studied, in which the fundamental period of the PIN and GAP were approximately 1.35 s for the primary bending mode (portal sway), while that of the FC is 1.36 s. Figure 3b shows the fundamental period of the PIN.

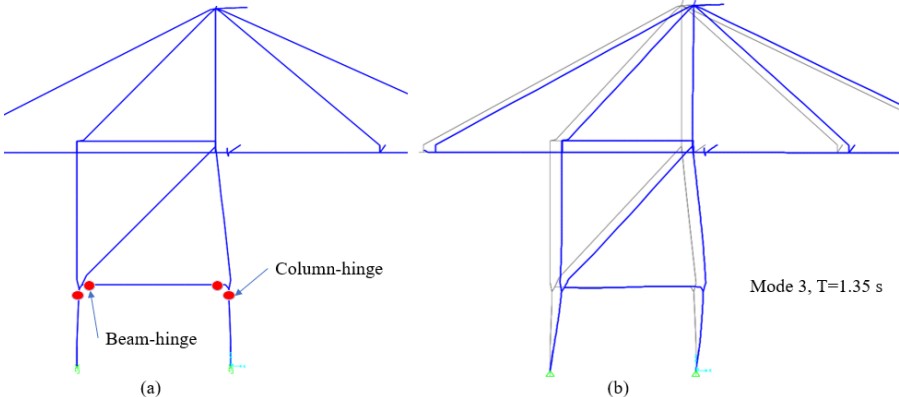

**Figure 3.** (**a**) Plastic hinge; (**b**) portal sway at the fundamental period *T* = 1.35 s.

### 2.3. Defining Damage Criteria for the Container Crane

In this study, as mentioned above, there were ten levels of $S_{ai}$ from 0.2 to 0.9 g to be analyzed. According to PIANC [25], the damage criteria for the container crane is divided into four levels: working within the elastic limit without derailment, working in the plastic limit with derailment, working within the plastic limit without collapse, and collapse. In Kosbab's thesis [18], research for container cranes also proposed four limit states in term of portal drift: Derailment, Immediate Use, Structural Damage, and Complete Collapse. In particular, Tran et al. [11] researched the same container crane type, and defined the performance level and damage criteria. The authors suggested three performance levels according to ASCE/SEI 41-13 [22]: Immediate occupancy (IO); Life safety (LS); and Collapse prevention (CP), as shown in Table 1. Using pushover analysis, the authors showed the portal drift limits corresponding to the three performance levels are 1.6, 1.8, and 2.2%, respectively, as shown in Figure 4. If the portal drift is larger than CP level, the structure may be serious damaged or collapse.

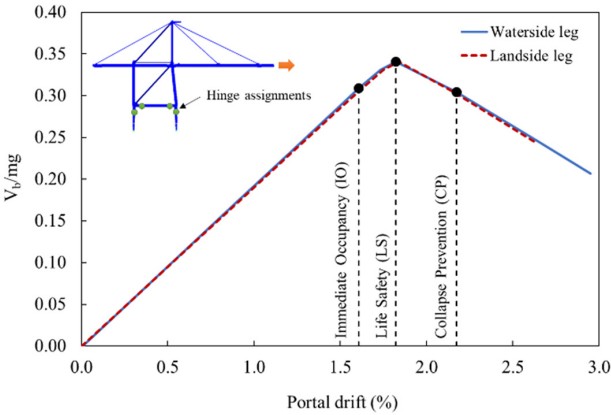

**Figure 4.** Pushover curve and limit states for the typical Korean container crane [11].

**Table 1.** Structural performance levels and damage [11].

| Elements | Immediate Occupancy (IO) | Life Safety (LS) | Collapse Prevention (CP) |
|---|---|---|---|
| For vertical elements of steel moment frame (i.e., a portal column of the crane) | Minor local yielding at a few places. No fractures. Minor buckling or observable permanent distortion of members. | Hinges form. Local buckling of some beam elements. Severe joint distortion, but shear connections remain intact. A few elements may experience partial fracture. | Extensive distortion of beams and column panels. Many fractures at moment connections, but shear connections remain intact. |
| Overall damage | Light | Moderate | Severe |

## 3. Recorded Ground Motions

Nine horizontal ground motions (GM), with magnitudes ranging 6.53 to 7.28, were selected for this study. Table 2 lists the information of earthquakes; there are seven ground motions for far-source stations, and two ground motions for near-source stations. Here, a near-source station is a station whose fault distance is less than 15 km. These data were obtained from the Pacific Earthquake Engineering Research Centre (PEER) and Huh et al. [27].

**Table 2.** Unscaled ground motions and spectral acceleration.

| No. (GM) | Earthquake | Year | Station Name | Mag. | $R_{rup}$ (km) | PGA (g) | $S_{ao}$ (g) ($T$ = 1.35 s) |
|---|---|---|---|---|---|---|---|
| 1 | Imperial Valley-02 | 1940 | Elcentro Array #09 | 6.95 | 6.09 | 0.28 | 0.23 |
| 2 | Imperial Valley-06 | 1979 | Elcentro Array #06 | 6.53 | 1.35 | 0.45 | 0.39 |
| 3 | Landers | 1992 | Barstow | 7.28 | 34.86 | 0.13 | 0.11 |
| 4 | Landers | 1992 | Yermo Fire Station | 7.28 | 23.62 | 0.24 | 0.53 |
| 5 | Loma Prieta | 1989 | Gilroy-Gavilan Coll. | 6.93 | 9.96 | 0.36 | 0.23 |
| 6 | Northridge-01 | 1994 | Newhall-Fire Stn. | 6.69 | 5.92 | 0.58 | 0.68 |
| 7 | Northridge-01 | 1994 | Rinaldi Receiving Stn. | 6.69 | 6.50 | 0.87 | 1.22 |
| 8 | Northridge-01 | 1994 | Sylmar-Converter Stn. | 6.69 | 5.35 | 0.62 | 0.80 |
| 9 | Kobe_Japan | 1995 | KJMA | 6.90 | 0.96 | 0.83 | 0.92 |

Note: $R_{rup}$ is the closest distance to the rupture plane (rupture distance).

The typical intensity measures (IM) of an earthquake are peak ground motions (acceleration, velocity, and displacement) and elastic response spectra (acceleration, velocity, and displacement), in which the elastic response spectra have some advantages of reflecting the behavior of structures under seismic excitation, and are often applied in the seismic analysis. In our study, the acceleration response spectra are selected to consider the response of the container crane when it has uplift and derailment phenomenon. A set of the ground motion profiles, in which each of them is scaled via

the ratio of target spectral accelerations ($S_{ai}$) and the original $S_{ao}$ of each ground motion, the $S_{ao}$ are determined at the fundamental period of the container crane ($T$ = 1.35 s). This technique was also used by Tran et al. [11], who compared two methods, the nonlinear static and time history methods, for a typical Korean STS container crane. The authors scaled the ground motions to the design earthquake with a return period of 2400 year for soil type $S_D$, according to the design response spectrum in KDS 2016 ($S_a$ = 0.21 g). This process to scale ground motions is simplified as follows:

- Compute the elastic spectral acceleration $S_{ao}$ at the fundamental period of the container crane ($T$ = 1.35 s) with damping 5% from original ground motions; Table 2 shows the $S_{ao}$.
- Determine the scaled factor ($\alpha_i$) by $\alpha_i = S_{ai}/S_{ao}$.
- The ground motions that were obtained times the original ground motion with $\alpha_i$ are used in the time history analysis.

Figure 5 shows the elastic response spectra with five-percent damping and the corresponding mean spectrum of nine horizontal time-history accelerations from the earthquakes. In this figure, the original spectral acceleration of each ground motion is determined at $T$ = 1.35 s, for example, the $S_{ao}$ of Northridge-01 (Rinaldi Station) earthquake at the fundamental period of the structure is 1.22 g. The figure also illustrates the elastic design spectra according to KDS 2016 [28] for stiff soil $S_D$ and seismic zone I (seismic zone factor = 0.22 g); this elastic design spectra is equivalent to a return period of 2400 years. The elastic design spectra are built for the site where the Korean container crane is located, and are used to evaluate the original ground motion. This determined the intensities of the target spectral acceleration, i.e., $S_{ai}$ = 0.2, 0.3, 0.35, 0.4, 0.45, 0.5, 0.55, 0.6, 0.8, 0.9 g; ten levels of $S_{ai}$ were taken into account to evaluate the responses of the container crane. In this study, the response of container crane was considered both uplift and no uplift behaviors. Figure 6 shows the scaled response spectra and design response spectrum.

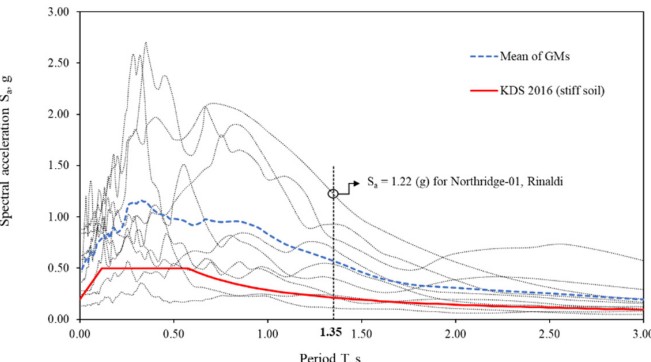

**Figure 5.** Elastic response spectra and design response spectra.

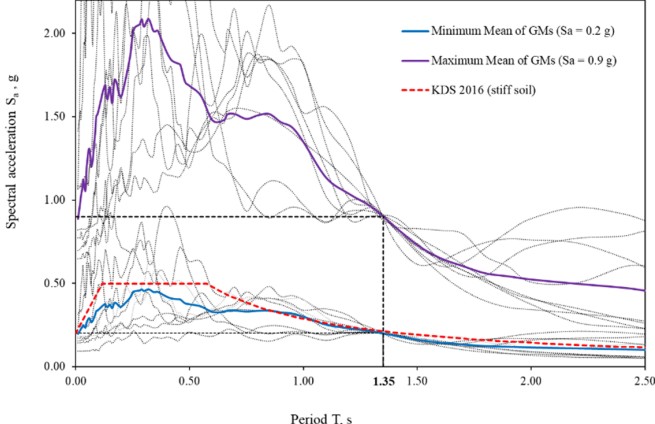

**Figure 6.** Scaled response spectra at the fundamental period.

The recorded ground motions include three components: two horizontal motions, and one vertical ground motion, so which components had the largest effect on the dynamic response of the crane needed to be considered. Jacobs et al. [29] investigated the effects of the different components of ground motions that were longitudinal excitation (applying in the trolley-travel direction) versus longitudinal plus horizontal excitation (applying in the rail direction), and the longitudinal only versus triaxial excitations (longitudinal, horizontal, and vertical excitations) on the response of the crane. For the longitudinal versus the longitudinal plus horizontal excitation, the comparative result showed that the difference of portal drift was very small (an increase or decrease), so it would be sufficient to consider only the longitudinal ground motions. For the longitudinal only and the triaxial excitations, there were large differences in the drift and vertical displacement; however, there was not a consistent trend of increase or decrease. In addition, the uplift did not influence whether the dynamic response of the crane was increased or reduced. Therefore, in this study, when analyzing the dynamics of the container crane, only the longitudinal excitation was considered to apply at base support in the trolley-travel direction (x-axis direction of Figure 1). Figure 7 shows the original longitudinal accelerations and a scaled longitudinal acceleration (e.g., for $S_a = 0.9$ g).

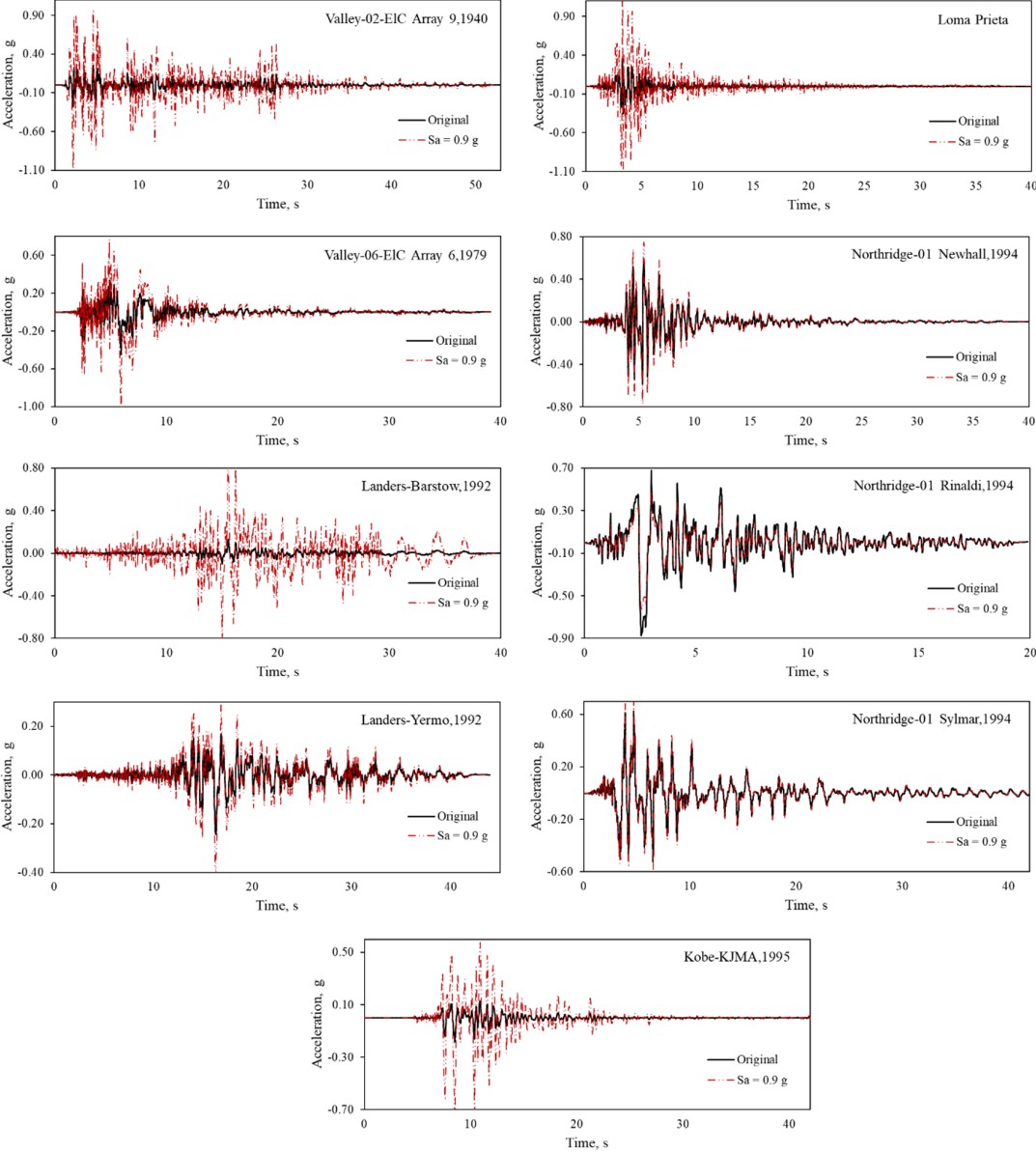

**Figure 7.** Original and scaled ground motions.

## 4. Results and Discussions

### 4.1. Displacement of Legs

The displacement of legs is only considered for GAP and FC models. The GAP boundary allows observation of vertical movement, while the FC boundary can be considered approximately in both vertical and horizontal movements. As mentioned above, the uplift event occurs when the vertical movement of the leg is positive, and its vertical force is equal to zero. Figure 8 shows for this analysis, the average value of uplift for each intensity of spectral acceleration for the landside leg (Node 10) of both FC and GAP elements.

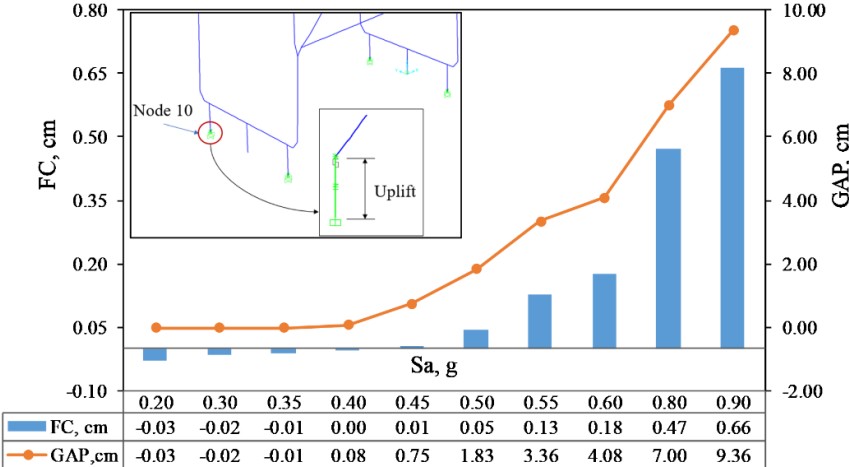

| Sa, g | 0.20 | 0.30 | 0.35 | 0.40 | 0.45 | 0.50 | 0.55 | 0.60 | 0.80 | 0.90 |
|---|---|---|---|---|---|---|---|---|---|---|
| FC, cm | -0.03 | -0.02 | -0.01 | 0.00 | 0.01 | 0.05 | 0.13 | 0.18 | 0.47 | 0.66 |
| GAP, cm | -0.03 | -0.02 | -0.01 | 0.08 | 0.75 | 1.83 | 3.36 | 4.08 | 7.00 | 9.36 |

**Figure 8.** Vertical displacement for the landside leg (Node 10).

The uplift event happened at a spectral acceleration of 0.45 g for the FC boundary, and its value of uplift was 0.01 cm, while the uplift event occurred early at a spectral acceleration of 0.4 g for the GAP boundary, and its value of uplift was 0.08 cm. When the spectral acceleration rose, the data for the GAP boundary increased gradually. The maximum of vertical movement under $S_a$ of 0.9 g for the FC boundary was 0.66 cm, while that for the GAP boundary was 9.36 cm.

Figure 9 shows the average horizontal movement of the landside leg (node 10) for the FC boundary. The derailment event firstly happened at a spectral acceleration of 0.45 g, and the derailment of node 10 was 8.2 cm. At the spectral acceleration of 0.9 g, the average derailment was 60.5 cm, whereas the minimum and maximum values are 41.3 cm and 75.1 cm for Northridge (Sylmar) and Imperial Valley (array 6), respectively.

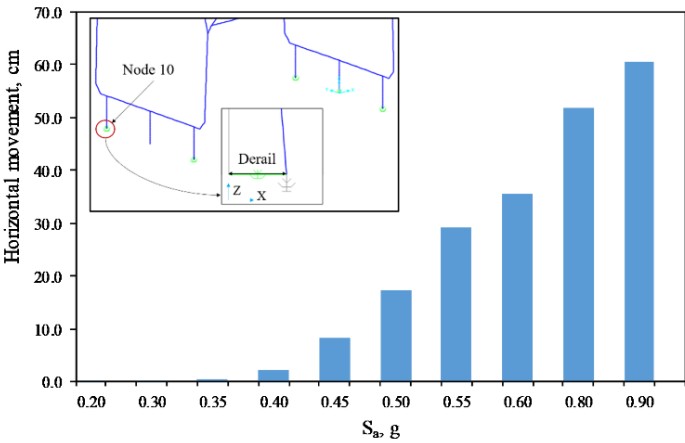

**Figure 9.** Horizontal movement of the landside leg (node 10) for the FC boundary.

## 4.2. Lateral Displacement of Portal Frames

In contrast to the analysis of buildings, where a horizontal displacement of the roof is often of interest, the dynamic analysis commonly considers the horizontal displacement or drift at the top of the portal frame of the container crane, because the drift provides information about how much the deformation in the fundamental mode occurs in the portal frame, which is the main structure supporting the entire structure above. Furthermore, during past earthquakes, most of the plastic hinges were recorded at the leg of the portal frame. In this study, four nodes of the portal frames, which are two seaside nodes (4052 and 3052) and two landside nodes (1052, 2052), were determined for the drift portal with the height of the portal frame being 17.5 m.

Figure 10 shows that for the PIN boundary, the portal drift of the seaside legs (average of two nodes 4052 and 3052) is plotted. However, in the real case, when the crane is operating, the crane's wheels can move freely on the rail. If the crane subjects a high impact of earthquake, it can be uplifted. Thus, the result obtained from the pin-support boundary should be verified with other boundary conditions. Figure 11 shows the portal drift in cases using the FC boundary. The results of PIN, GAP, and FC boundaries are compared in Figure 12.

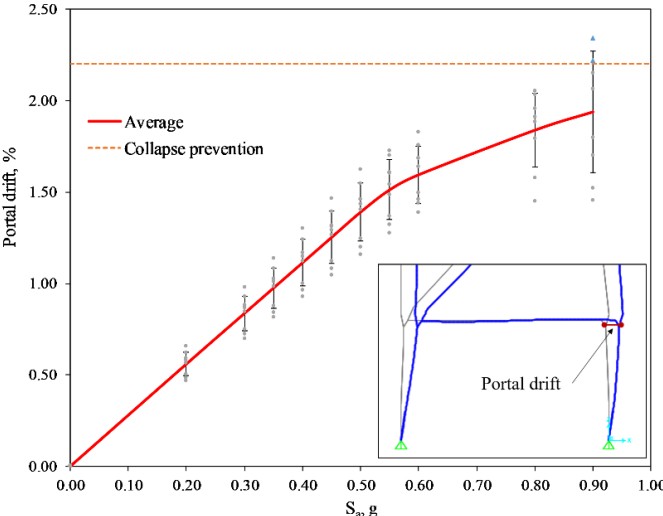

**Figure 10.** The portal drift of the PIN boundary condition.

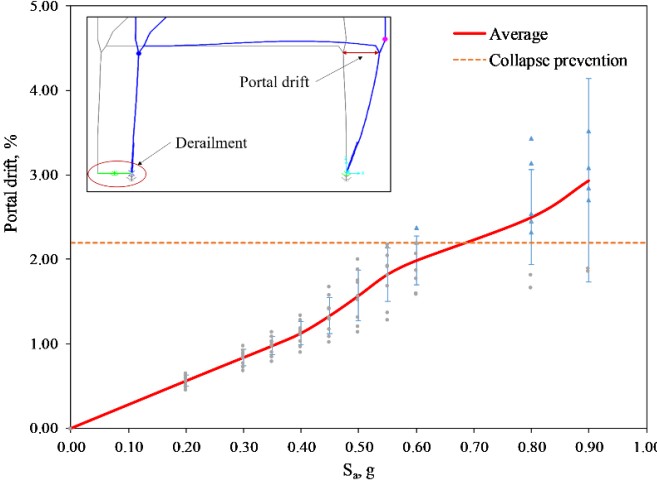

**Figure 11.** The portal drift of the FC boundary condition.

Figure 12 shows the difference of the portal drift for other boundaries. When the spectral acceleration value is less than 0.4 g, there is no difference in using the simulated boundaries, and the

relationship between spectral acceleration and horizontal displacement is linear. However, when the spectral value is greater than 0.4 g, using the simulated boundary model affects the result of the portal drift, and the relationship is now nonlinear. The FC boundary has the largest horizontal displacement, as opposed to the smallest of the GAP boundary. This could be explained by the influence of the uplift and derailment phenomenon; while the FC boundary allows uplift and derailment to occur, the GAP boundary only allows the uplift phenomenon, and the PIN boundary does not allow both uplift and derailment. Under increasing IM, the increasing trend of PIN and GAP boundaries gradually decrease with a smaller slope than in the linear part, while the trend still increases in the case of the FC boundary because the horizontal movement of the legs is allowable.

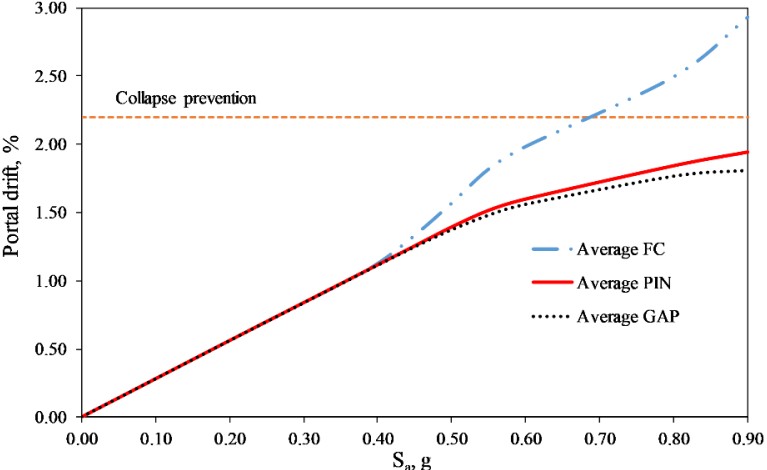

**Figure 12.** The portal drift of the FC boundary condition.

### 4.3. Reaction of Legs

Consider the reaction of the landside legs in which uplift and derailment events happened first. Figure 13 shows the result of horizontal reaction of the landside leg (node 10) at the spectral acceleration of 0.2 and 0.6 g of the Imperial Valley-02 earthquake. When the intensity of earthquake is low ($S_a = 0.2$ g), there is almost no difference between the boundary condition models. The maximum difference is only about 60.54 kN at 25.06 s for FC vs. PIN, while that for GAP vs. PIN is approximately 45.85 kN at 25.07 s, and FC vs. GAP is 15.48 kN at 19.61 s. As compared to the maximum values of horizontal reaction, which are 377 and 384 kN for FC and PIN, respectively, the difference determined as percentage is 15.8% for the couple of FC vs. PIN, while that for GAP vs. PIN is 6.5%, while FC vs. GAP is 4.11%.

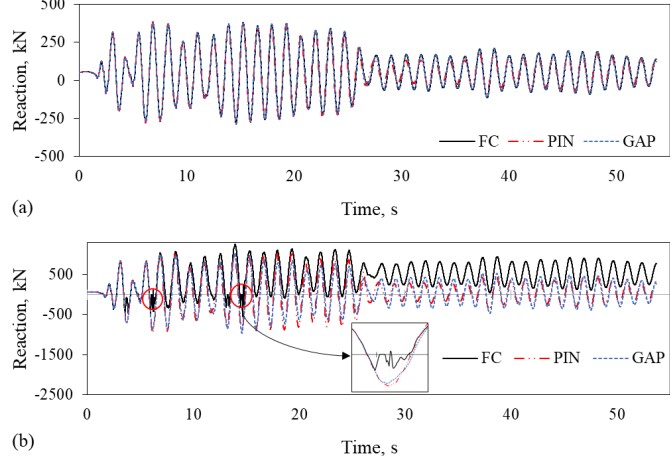

**Figure 13.** Horizontal reaction of landside leg: (**a**) at $S_a = 0.2$ g; (**b**) at $S_a = 0.6$ g.

With regard to Figure 13b, the intensity of spectral acceleration of 0.6 g—in which uplift and derailment events occurred—when using the different boundary condition models, there is overall a fluctuation in the horizontal reaction of the landside legs. At the beginning time, the initial equilibrium position indicates a horizontal reaction of around 53.74 kN on the landside leg because of the effect of gravity. When the landside leg is derailed, its reaction profile shows different behavior, as illustrated in Figure 13b. The maximum differences are 1024.75 kN at 6.21 s, 402.75 kN at 18.37 s, and 969.90 kN at 14.6 s for FC vs. PIN, GAP vs. PIN, and FC vs. GAP, respectively. As compared to the maximum value of horizontal reaction at $S_a$ = 0.6 g, the values of FC and PIN are 1252 and 1045 kN, respectively, and the difference determined as percentage is high at 81.8% for the couple of FC vs. PIN, while that of GAP vs. PIN is 38.5%, and FC vs. GAP is 77.4%. Notably, the horizontal reaction of the landside legs in the FC boundary model is equal to zero when the uplift event occurs, while the value of landside legs for GAP and PIN does not equal zero, as in Figure 13b, which indicates that there are two periods in which the uplift event happened, 6.05 to 6.13 s, and 14.54 to 14.65 s (as can be seen the red circles of Figure 13b).

Figure 14 shows the average total base shear obtained from the time history analyses. In this data, there are no differences in the total base shear for the other boundary models when the spectral acceleration is less than 0.4 g, and the relationship between total base shear and spectral acceleration is linear. However, the difference begins after the spectral acceleration of 0.4 g as a result of the phenomenon of uplift and derailment of the container crane, and this relationship is no longer linear. This could be explained by the effect of the boundary condition model, as the PIN boundary does not reflect the correct working condition; it fixes the legs into the ground (rail), and then the resistance of the link makes the total base shear increase significantly. In contrast, the GAP and FC boundary allow uplift or both uplift and derailment, and the effect of the free movement of the legs results in the relationship also becoming nonlinear. In the three boundary models, when uplift and derailment happen, the PIN boundary has the largest total base shear, while the FC boundary has the lowest one. Throughout, the distinction of the dynamic response of the container crane between the bonded contact model (PIN) and unbonded contact (FC or GAP) begins during the uplift and derailment event, $S_a$ = 0.4 g.

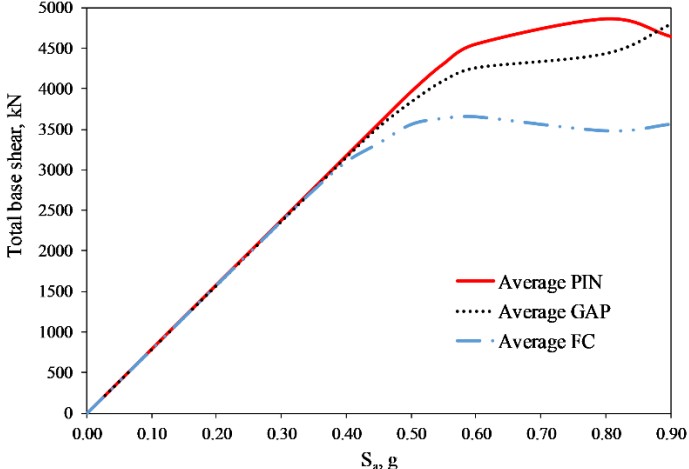

**Figure 14.** Total base shear.

## 5. Conclusions

This study presented the effect of boundary condition models on the response of container cranes under seismic load. The research took three boundary condition models into account for a 3D model of the Korean container crane, and used nine real recorded ground motions to evaluate the dynamic behavior of the structure by time history analysis. The following are the key conclusions obtained from this study.

When the intensity of an earthquake is low (i.e., the $S_a$ is less than 0.4 g at the fundamental period of the container crane), the boundary condition of the container crane model can be applied by PIN, GAP element, or Friction contact, which does not significantly affect the response of the structure, as measured in terms of the portal drift, movement, and reaction of the legs. In this case, the uplift and derailment events are supposed to not happen, or only occur slightly, so that the dynamic response of the container crane under seismic excitation is not affected by the boundary conditions.

When the intensity of earthquake is high enough, which can create uplift and cause a derailment event, the boundary condition of the container crane model needs to be considered in the analysis. If only considering a pure uplift event, the GAP boundary could be suitable for the analysis. However, the response of the container crane under high seismic condition is both uplift and derailment, which happen simultaneously. The boundary condition by Friction contact could approximately express both the uplift and derailment phenomena. The PIN boundary should not be applied in this case, because it increases the total base shear and reduces the portal drift. In particular, it can't express the widening of the container crane legs, which is crucially important in analyzing the container crane structure.

**Author Contributions:** Conceptualization, J.H., V.B.N. and C.K.; Methodology, J.H.; Software, V.B.N. and Q.H.T.; Validation, J.H.A. and C.K.; Formal Analysis, V.B.N. and Q.H.T.; Resources, J.H.; Data Curation, V.B.N. and Q.H.T.; Writing-Original Draft Preparation, V.B.N. and C.K.; Writing-Review and Editing, J.H. and J.H.A.; Supervision, J.H. and C.K.

**Funding:** This research was funded by [Ministry of Oceans and Fisheries, Korea] and [National Research Foundation of Korea] grant number [2017R1D1A3B03032854], and the APC was funded by [National Research Foundation of Korea].

**Acknowledgments:** This research was a part of the project entitled 'Development of performance-based seismic design technologies for advancement in design codes for port structures', funded by the Ministry of Oceans and Fisheries, Korea and was also supported by the Basic Science Research Program through the National Research Foundation of Korea (NRF), funded by the Ministry of Education (2017R1D1A3B03032854).

**Conflicts of Interest:** The authors declare no conflict of interest.

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
