# Peer review of "Effects of Boundary Condition Models on the Seismic Responses of a Container Crane"

_applsci, doi:10.3390/app9020241_

Round 1

Reviewer 1 Report

The following comments and suggestions can give more clarity to the research presented.

Review comments

1.      An explanation of equation (4) should be given. How the authors derived it?

2.      In a Figure containing the crane the authors should show the three components of motion they mention at the top of page 9.

Reviewer 2 Report

In this manuscript, the dynamic response of container cranes under earthquake action considering different boundary conditions is analysed. The paper is well-written and organized in general. The results of the research may be useful for designers of these type of structures. Nevertheless, the manuscript has two main limitations: (i) it is not clear for the reviewer the contribution of the manuscript again the previous contribution (this point should be highlighted in the introduction of the manuscript) and (ii) the validation of the results has been performed only numerically. The first limitation should be clarified in the manuscript in order to be considered for publication.
